# Direct Reprogramming and Induction of Human Dermal Fibroblasts to Differentiate into iPS-Derived Nucleus Pulposus-like Cells in 3D Culture

**DOI:** 10.3390/ijms23074059

**Published:** 2022-04-06

**Authors:** Shoji Seki, Mami Iwasaki, Hiroto Makino, Yasuhito Yahara, Yoshitaka Miyazaki, Katsuhiko Kamei, Hayato Futakawa, Makiko Nogami, Nguyen Tran Canh Tung, Tatsuro Hirokawa, Mamiko Tsuji, Yoshiharu Kawaguchi

**Affiliations:** 1Department of Orthopaedic Surgery, Faculty of Medicine, University of Toyama, Toyama 930-0194, Japan; mhigashi@med.u-toyama.ac.jp (M.I.); hiroto@med.u-toyama.ac.jp (H.M.); k.kamei9786@gmail.com (K.K.); hayato_care@yahoo.co.jp (H.F.); makohna@hotmail.com (M.N.); bstungbv103@gmail.com (N.T.C.T.); tatsuro.h13@gmail.com (T.H.); tsujimam@med.u-toyama.ac.jp (M.T.); zenji@med.u-toyama.ac.jp (Y.K.); 2Department of Molecular and Medical Pharmacology, Faculty of Medicine, University of Toyama, Toyama 930-0194, Japan; yasuhito.yahara@hotmail.co.jp; 3KUBIX Inc., ha8-1 Netsuno, Hakusan 920-2161, Japan; yoshitaka.miyazaki@kubix.co.jp; 4Department of Trauma and Orthopaedic Surgery, Vietnam Military Medical University, Hanoi 100000, Vietnam

**Keywords:** intervertebral disc disease, human dermal fibroblast, nucleus pulposus cell, alginate beads, direct reprogramming, intervertebral disc regeneration, induced pluripotent stem cells

## Abstract

Intervertebral disc (IVD) diseases are common spinal disorders that cause neck or back pain in the presence or absence of an underlying neurological disorder. IVD diseases develop on the basis of degeneration, and there are no established treatments for degeneration. IVD diseases may therefore represent a candidate for the application of regenerative medicine, potentially employing normal human dermal fibroblasts (NHDFs) induced to differentiate into nucleus pulposus (NP) cells. Here, we used a three-dimensional culture system to demonstrate that ectopic expression of *MYC*, *KLF4*, *NOTO*, *SOX5*, *SOX6*, and *SOX9* in NHDFs generated NP-like cells, detected using Safranin-O staining. Quantitative PCR, microarray analysis, and fluorescence-activated cell sorting revealed that the induced NP cells exhibited a fully differentiated phenotype. These findings may significantly contribute to the development of effective strategies for treating IVD diseases.

## 1. Introduction

Intervertebral disc (IVD) disease is a common spinal disorder, which causes neck or back pain in the presence or absence of an underlying neurological disorder. Spinal fusion surgery, artificial disc arthroplasty, and spinal-canal decompression surgery are employed worldwide to treat IVD diseases [1,2,3,4,5]. Spinal fusion is not preferred because it does not preserve intervertebral mobility. Furthermore, the numerous disadvantages of artificial disc arthroplasty include rupture of the artificial disc, vulnerability to infection, and difficulty in replacing an artificial disc [6]. Therefore, alternative therapeutic strategies are urgently required.

Disc degeneration characterizes the first stage of IVD diseases, leading to disc herniation, spinal stenosis, and discogenic lower back pain. Disc degeneration occurs during aging, and established treatments are not available to prevent this pathology. During the last few decades, clinical trials and animal experiments designed to treat IVD diseases include stem cell therapy [7], artificial disc transplantation [8], and the administration of catabolic factors [9]. The results of these efforts indicate that disc regeneration represents a promising therapeutic approach with the potential to confer significant benefits.

The IVD comprises the nucleus pulposus (NP), the annulus fibrosus (AF), and the endplate. An endplate of IVD is the transition region where a vertebral body and intervertebral disc interface with each other, and the endplate is divided into two parts, cartilaginous and bony endplate. The cartilaginous endplate contains cartilage-like cells and helps maintain the form and function of the disc. The cartilage-like NP contains an abundant extracellular matrix (ECM) consisting of proteoglycans, aggrecan, and collagens [10]. Cellular reprogramming generates alternative cell types through the overexpression of transcription factors that determine target-cell fate. For example, expression of *Oct3/4*, *Sox2*, *c-Myc*, and *Klf4* reprograms mature fibroblast to acquire a pluripotent phenotype similar to that of an embryonic stem cell [11,12]. Furthermore, direct reprogramming of normal human dermal fibroblasts (NHDFs) to differentiate into chondrogenic cells is accomplished using ectopic expression of genes encoding differentiation factors such as *c-Myc*, *Klf4*, and *Sox9* [13]. For example, the reprogramming factors *c-Myc* and *Klf4* are necessary and sufficient to differentiate NHDFs into chondrogenic cells without differentiating through a pluripotent stem cell phenotype [14]. Furthermore, *SOX9* induces chondrocyte differentiation by increasing the expression levels of chondrocyte-specific type II collagen (*COL2A1*) [15,16]. Moreover, *SOX9* is required for axial skeletogenesis through maintaining the notochord and chondrogenesis [17].

*SOX5* and *SOX6* are required for the differentiation, maturation, and survival of the NP. For example, *SOX5*, *6*, and *9* (*SOXtrio*) cooperate to induce the differentiation of target cells into chondrogenic cells [18,19]. *SOX5* and *SOX6* drive the expression of the gene encoding aggrecan (*ACAN*) in cooperation with *SOX9* [20], and they induce the development of the nucleus pulposus of IVDs [20]. Furthermore, *SOX5*^−/−^ and *SOX6*^−/−^ mice are unable to form the notochord sheath [21]. The *NOTO* transcription factor induces pluripotent stem cells to differentiate along the notochordal lineage [22], and studies of *Noto*-cre mice show that notochordal cells serve as tissue-specific stem cells within the IVD during growth, culminating in their differentiation to form the NP [23].

Moreover, these finding suggests that NOTO is required during the early stage of differentiation into NP cells. However, we are unaware of reports describing the use of defined differentiation factors that directly reprogram and induce the differentiation of NHDFs to NP-like cells. Therefore, the present study aims to fill this gap in our knowledge, with the ultimate goal of establishing an effective treatment strategy for IVD diseases.

## 2. Results

We performed transcriptional profiling, immunofluorescence analysis, microarray, and histological analyses of mock-transfected NHDFs and the *SOX9*- and *SOXtrio*-transfected groups (collectively referred to below as the NHDFs/NHDF groups described in Section 4).

### 2.1. Transcriptional Analysis of NHDFs

Figure 1 shows the results of qPCR analysis of transcripts encoding ECM proteins, differentiation factors, and matrix metalloproteinases. The expression levels of *ACAN*, which was the most abundantly expressed component of the IVD, were significantly increased in the NOTO-SOX9 and NOTO-SOXtrio groups. *COL2A1* and *COL9A2* mRNAs, encoding cartilage-specific collagens, were significantly increased in both groups. *COL11A2* was also significantly increased in NOTO-SOXtrio groups. At the same time, *ADAMTS4* was significantly increased in the NOTO-SOX9 and NOTO-SOXtrio groups. *CD24* was significantly increased in NOTO-SOXtrio groups compared with that of the NOTO-SOX9 group. The levels of *GLI3* and *GD2* mRNAs, which serve as markers for the degree of cell differentiation in the IVD, were significantly increased, and those of *TEK*, which serve as markers for the degree of cell differentiation in the IVD, were significantly decreased.

### 2.2. Analysis of Glycosaminoglyan (CAG) Expression in NHDFs

Figure 2 shows that hematoxylin and eosin (HE) staining revealed more cells with an acidophilic cytoplasm in the NOTO-SOXtrio goup compared with those of the NOTO-SOX9 and control groups, and Safranin-O staining revealed small round cells in the control and experimental groups. Furthermore, the NOTO-SOXtrio group exhibited more intense metachromasia among the groups, suggesting that the former produced the highest levels of GAGs.

### 2.3. Immunofluorescence Analysis of the Expression of ECM Components in Cultures of NHDFs

Immunofluorescence revealed that Aggrecan and Type II collagen, which are characteristic of the NP, were abundantly expressed in marker-gene transfected NHDFs but not in control transfectants. The NOTO-SOX9trio group expressed the highest level of each ECM component, which is mainly localized to the extracellular space (Figure 3).

### 2.4. Detection of Vacuolated Cells in Cultures of the NOTO-SOXtrio Group

NP cells, which are characteristically vacuolated, form fibroblastic colonies (CFU-F) and colony-forming spherical cells (CFU-S) [24]. Such cells observed here are referred to as “NP-like”. Figure 4 shows that compared with controls (Figure 4a), vacuolated cells and intracellular vesicles were present in cultures of the NOTO-SOXtrio group (Figure 4b). Furthermore, the NOTO-SOXtrio group formed CFU-S (Figure 4b). These properties may be associated with abundant expression of ACAN and type II collagen as well as with the expression of GD2/TEK-double-positive cells, as previously reported [24]. In contrast, CFU-S were not detected in the control group (Figure 4a).

### 2.5. Genome-Wide Microarray Analysis

We isolated human NP cells from patients with scoliosis, some of which were cultured with alginate beads for 4 weeks to serve as a positive control. Figure 5a shows a heatmap analysis of 22,425 genes included in the microarray. Clustering analysis indicated similarity between the transcriptional profiles of the NOTO-SOX9 and NOTO-SOXtrio groups. Scatter plot analysis of the NOTO-SOXtrio and positive control (human NP cells) showed similar gene expression patterns. The number of genes with expression levels differencing by 2-fold between the human NP cells and the NOTO-SOXtrio was 17,643 of 22,425.

We next analyzed the expression of NP-marker genes (54 out of 22,425 genes) [25]. Heatmap analysis revealed similar expression patterns between the two groups (Figure 5c). Scatter plot analysis showed that the expression levels of 44 out of 54 NP-marker genes were within two-fold between the human NP cells and the NOTO-SOXtrio group, and those of the remaining 10 were within three-fold (Figure 5d).

### 2.6. Fluorescence-Activated Cell Sorting (FACS) Analysis to Determine the State of Differentiation of NP-like Cells

The phenotypes of NP-like cells in the NOTO-SOXtrio group and human NP cells were determined using FACS (Figure 6). Double-positive (CD24+/GD2+) cells were more abundant in the NOTO-SOXtrio group compared with human NP cells as a positive control. In contrast, CD24+/TEK+ and GD2+/TEK+ cells were more abundant in the human NP cell culture compared with those of the NOTO-SOXtrio group. The expression patterns of differentiation markers were similar between the two groups, indicating that the NP-like cells in the NOTO-SOXtrio group exhibited a differentiated phenotype.

## 3. Discussion

Here, we used 3D cultures of NHDFs transfected with vectors expressing differentiation factors to show the direct reprogramming and differentiation of NHDFs into NP-like cells. Furthermore, this 3D culture method employing alginate beads is required for propagating and maintaining IVD cells, particularly NP cells, to produce abundant ECM proteins such as Aggrecan and type II collagen [10]. We found that Safranin-O staining identified the NOTO-SOXtrio group as the most abundant producer of GAGs. Transcriptional analysis showed that *ACAN* and *COL2A1* levels were highest in the NOTO-SOXtrio group. Moreover, microarray and FACS results were consistent regarding the expression of NP-specific genes and differentiation makers. These findings indicate that the NOTO-SOXtrio group exhibited the most highly differentiated phenotype among the three NHDF groups.

The NOTO transcription factor is required for the differentiation of a mesodermal progenitor into axial mesodermal and notochordal fates, which is regulated by upstream signaling through the WNT and NODAL signaling pathways [23]. Furthermore, lineage-tracing experiments using Noto-cre mice show that the notochord arises from the NP when the IVD fully forms, preceding the formation of the axial skeleton [22,26]. Other candidate proteins that regulate differentiation of the early notochord include Shh, T (Brachyury), and Foxa2 [26,27]. However, we were unable to demonstrate that these molecules drove the differentiation of NP cells in our 3D culture system.

Sox9 regulates chondrogenesis [16], and heterozygous mutations at the SOX9 locus cause the autosomal dominant disease campomelic dysplasia of humans [28]. Furthermore, Sox9 conditional knockout mice exhibit severe IVD degeneration and reduction of vertebral size [29]. Moreover, Sox9 is required for the survival and maintenance of IVD cells [29]. Sox5 and Sox6, which are co-expressed with Sox9 during development, regulate the action of the latter [18,19]. These factors are required for notochord homeostasis through the formation of a notochord sheath, which enhances the production of ECM proteins such as Aggrecan and Type II collagen [18,19]. These findings suggest that the combination of differentiation markers ectopically expressed in NOTO-SOXtrio cells was required for the differentiation of NHDFs into mature NP cells.

Transcriptional profiling of NP-like cells detected the expression of the early notochord markers Gli3, Cd24, Tek, and Gd2 in the NOTO-SOX9 and NOTO-SOXtrio groups [25,26,27]. Gli3 functions as a downstream mediator of Shh and Ihh signaling and is required for Shh-dependent sclerotome induction [30]. Gli3 is expressed in mature bovine NP cells [31]. Furthermore, the early mouse notochord is characterized by the expression of transcription factors essential for its development, such as T (Brachyury) [32,33] or Foxa2 [34]. Here, we detected low levels of their respective transcripts.

Microarray analysis of NP-specific markers detected low levels of early notochord-specific markers, although CD24 encodes a cell surface NP-specific marker [35]. Furthermore, CD24, which is not expressed in chondrosarcomas, is detected in the herniated NP and chordomas and is therefore considered a surface marker specifically expressed by NP cells [35]. TEK and GD2 expression identify populations of NP progenitor cells in humans and mice [36], and the patterns of CD24, TEK, and GD2 expression change depending on the degree of cellular differentiation [36]. For example, TEK is expressed during the early stage of differentiation of the NP progenitor cell, after which its levels decline. The expression of CD24 increases during the differentiation of the NP (NP-committed cells [36]), and the expression of GD2 increases during the differentiation of NP progenitor cells to NP-committed cells and decreases during the final stages of differentiation [36]. Here, we show that these expression patterns did not significantly differ between normal NP cells and the NOTO-SOXtrio group. Furthermore, the expression levels of GD2 and CD24 in the NOTO-SOXtrio group were significantly higher compared with those in the NOTO-SOX9 group, while TEK expression levels were significantly lower compared with those of the control. Therefore, we suggest that the NOTO-SOXtrio group may have differentiated from NP progenitor cells into NP-committed cells. These findings further indicate that the NOTO-SOXtrio phenotype represents well-differentiated NP cells.

We show here that vacuolated and colony-forming cells were present in cultures containing alginate beads incubated in a chondrogenic medium containing transforming growth factor (TGF)-β1. Three-dimensional cultures of NP cells more efficiently preserve TEK+ cell populations and increase chondrogenic differentiation potential compared with 2D expansion [37]. For example, TEK+ cells maintain the expression of multipotent genes expressions in 3D cultures containing alginate beads [37]. According to a confocal microscopy analysis of alginate-bead cultures, NP cells cultured on 2D-plastic or -glass substrates rapidly lose their phenotype and consequently alter their gene and protein expression patterns [38]. On the other hand, highly mature and undegenerated human NP cells may not have so many vacuolated and colony-forming cells. Although we believe that the NP cells of the 3D culture were able to maintain the phenotype, it may have been better to perform the culture in a hypoxic compartment in order to better mimic the milieu of the IVD. This is our limitation in this study.

In contrast, TGF-β signaling is required for the development of the spine and disc tissue during embryonic development [39]. Analysis of TGF-β receptor knockout mice shows that TGF-β signaling is required for maintaining the growth plate and endplate during postnatal development. Lack of TGF-β signaling in growth plate chondrocytes and inner AF cells leads to loss of the ECM and endplate cartilage cells, which is accompanied by abnormal morphology of the growth plate cartilage [40]. Furthermore, TGF-β1 confers a differentiated phenotype upon NP-like cells, which serve as the basis of molecular therapies employing intradiscal injection [41], bone marrow mesenchymal stem cells [42,43], stem cells-derived cartilage endplate cells [44], or adipocytes [45]. Thus, TGF-β1 effectively differentiates diverse cell populations into NP-like cells [44]. Thus, 3D culture with alginate beads combined with TGF-β signaling is required for directly differentiating and reprogramming NHDFs to generate NP-like cells in vitro. Additionally, considering potential targets of sources for cell therapy as described above (Figure 7), induced pluripotent stem cells [23,24], adipose-derived or bone marrow-derived mesenchymal stem cells [42,43,46], stem cells-derived cartilage endplate cells [44], and adipocytes [45] have been reported. Here, we found that NHDF differentiates into NP-like cells by inducing defined factors. It may be a new potential target of cell sources for the treatment of IVD diseases in the future.

Finally, there is no evidence that these transplanted NP-like cells survive and engraft into the host tissue within the inhospitable degenerative disc environment. Biomaterial scaffolds can provide mechanical and structural support in the regeneration of IVDs and may also serve as a carrier for therapeutic cell delivery. An essential consideration of the cell therapy technique is the scaffolds in which cells are implanted. Although various scaffolds have been tried in clinical or animal models, such as collagen sponge [47], collagen microspheres [48], fibrin [49], hyaluronic acid hydrogels [50], and alginate beads [51], it has not reached the general treatment for IVD diseases. Gels and scaffolds can enhance in situ incorporation and guide differentiation. Scaffolds themselves are a potential method for encapsulation, preventing cell leakage. Further cellular transplantation and engraftation techniques are needed in the future.

## 4. Materials and Methods

### 4.1. Transfection and Construction of Expression Vectors

NHDFs from adult donors were obtained from PromoCell (Heidelberg, Germany). Plasmid DNAs encoding human genes used for transfection (MYC: OHu27105D, KLF4: OHu26293D, NOTO: OHu30602D, SOX9: OHu19789D, SOX5: OHu20243D, SOX6: OHu12492D) were obtained from GenScript (Piscataway, NJ, USA). The OHu number is indicated by the GeneScript Clone ID number. Expression constructs were used to transform Escherichia coli DH5α (TOYOBO, Osaka, Japan). DNA extraction was performed using a HiSpeed Plasmid Maxi Kit (Qiagen, Valencia, CA, USA). DNA concentrations were measured using a Nano Drop One spectrophotometer (Thermo Fisher Scientific, Waltham, MA, USA). All experiments were performed according to each manufacturer’s protocol. The culture conditions and reagents are the same as those described above. NHDFs (3.5 *×* 10^5^ cells) were added to 10 cm dishes 16 h before transfection. Transfection was performed using TransfeX (ATCC, Manassas, VA, USA), following the manufacturer’s instructions. Briefly, 1 mL of Opti-MEM I Reduced-Serum Media and human plasmid DNA (10 µg) and TransfeX (20 µg) were mixed and incubated for 15 min at room temperature. The culture medium was exchanged for a fresh medium before transfection. The TransfeX:DNA complexes were added to dishes (1 mL/dish), which were gently and reciprocally rocked. The transfected cells were incubated for 24 h in a humidified incubator at 37 °C in an atmosphere containing 5% CO_2_ for 24 h, followed by a medium exchange. Controls (empty pcDNA3.1-transfected) and cells transfected with expression vectors encoding differentiation markers designated NOTO-SOX9 (MYC, KLF4, NOTO, and SOX9 and NOTO-SOXtrio (MYC, KLF4, NOTO, SOX5, SOX6, and SOX9).

### 4.2. Alginate Beads Culture

Alginate beads were fabricated using the needle extrusion method with a 1.2% (*w*/*v*) solution of sodium alginate from brown algae (Sigma-Aldrich, MO, USA) was prepared. Alginate powder was dissolved in autoclaved 150 mM NaCl. Adherent, transfected NHDFs were removed from the surface culture dish using 0.25% trypsin-EDTA. The alginate beads were suspended with the transfected NHDFs (6 × 10^6^ cells/mL) and passed through an 18-gauge needle into chilled 150 mM NaCl containing 102 mM CaCl_2_. The resultant alginate beads were completely polymerized by incubation in the CaCl_2_ solution for 15 min and then washed twice with basal Dulbecco’s Modified Eagle Medium (DMEM). The beads were then transferred to a 6 cm petri dish containing Chondrocyte Differentiation Medium (CC-3225; Lonza, Basel, Switzerland) and cultured at 37 °C in a humidified atmosphere containing 5% CO_2_ for 1 week for an analysis of gene expression and for 4 weeks for histological analysis. Chondrocyte Differentiation Medium was supplemented with R3-IGF, TGF-beta, insulin, GA-1000, transferrin, and fetal bovine serum (FBS). The medium was replaced once each week. To isolate RNA from cells encapsulated in the alginate beads, 55 mM sodium citrate containing 150 mM NaCl was added to the dishes, which were gently shaken for 10–15 min to completely dissolve the beads.

### 4.3. Isolation of Human NP

IVD tissue was obtained from a patient with scoliosis who underwent surgery at Toyama University Hospital. Three scoliotic patients were included in this study. The intervertebral discs of 6–8-year-old boys with congenital scoliosis were removed and used for this study. All patients had nothing of IVD degeneration confirmed by MRI. The Ethics Review Committee of our institution approved the study (approval number: I2013004). Written informed consent was obtained from all patients before the collection of specimens. The NP was cut into small fragments and digested with collagenase. The cells were then filtered through a nylon mesh (pore diameter, 70 μm) (Corning, AZ, USA), centrifuged at 1600 rpm, and the supernatant was discarded. The pellet was washed twice with phosphate-buffered saline (PBS). The cells were cultured in a 10 cm dish containing (DMEM) under consistent culture conditions (37 °C, 5% CO_2_). Medium, which was supplemented with 10% FBS and 1% antibiotics (100 U/mL penicillin, 100 mg/mL streptomycin), was replaced every week. For subcultivation, the cells were detached from the plates with trypsin/EDTA and expanded. NP cells were subcultured 2 or 3 times to increase the number of cells. We try to keep the cells used for the experiment within 3 passages in total, including subculture. If the phenotype of NP cells was dedifferentiated within 3 passages, the cells were basically destroyed or redifferentiated with alginate beads with Chondrocyte Differentiation Medium (CC-3225; Lonza, Basel, Switzerland) for experimental study.

### 4.4. qPCR

Total RNA was prepared from transfected NHDFs using Isogen (Nippon Gene, Tokyo, Japan). Extraction of RNA extraction was performed using a PureLink RNA Mini Kit (Thermo Fisher Scientific, Waltham, MA, USA), and cDNAs were synthesized using a High-Capacity RNA-to-cDNA Kit (Thermo Fisher Scientific, Waltham, MA, USA) with a thermal cycler. Extracted RNA was measured using a Nano Drop One spectrophotometer (Thermo Fisher Scientific, Waltham, MA, USA). Gene expression analysis (qPCR) was performed using the iTaq Universal SYBR Green Supermix (Bio-Rad, Hercules, CA, USA) with a CFX Connect (Bio-Rad, Hercules, CA, USA). All experiments were performed according to each manufacturer’s protocol. The RNA expression levels were normalized to those of GAPDH mRNA.

### 4.5. Hematoxylin–Eosin and Safranin-O Staining and Immunofluorescence Analysis

For histological analysis, the beads were fixed in 4% paraformaldehyde, 0.1 M sodium-cacodylate buffer containing 20 mM CaCl_2_ for 4 h at 4 °C, and then washed overnight at 4 °C in 0.1 M cacodylate buffer containing 20 mM CaCl_2_. CaCl_2_ was added to prevent the disintegration of the beads. The beads were dehydrated in a graded series of methanol concentrations, washed in xylene, and embedded in paraffin. Sections (5 μm thick) were cut using a microtome (IVS-410, Sakura Finetek Japan, Tokyo, Japan) and mounted on glass slides. Deparaffinized sections were stained with hematoxylin–eosin (131-09665, FUJIFIRUM Wako, Osaka, Japan), eosin Y solution (051-06151, FUJIFIRUM Wako, Osaka, Japan), and Safranin-O (Safranin; 4216-1, MUTO Pure chemicals, Tokyo, Japan, Fast green; 069-00032, FUJIFIRUM Wako, Osaka, Japan) to analyze) to analyze cartilage tissue formation. For immunofluorescence analysis, the deparaffinized sections were permeabilized with 0.1% Triton X-100 (065K0122, Merck, Darmstadt, Germany) for 30 min, washed twice with PBS, and then incubated for 1 h at room temperature in 3% BSA in PBS (blocking solution). The cells were then incubated overnight at 4 °C with rabbit polyclonal anti-type II collagen (ab34712, Abcam, Cambridge, UK) and mouse monoclonal anti-aggrecan (sc-33695, Santa Cruz, TX, USA) antibodies (diluted 1:50 in 1% BSA solution). The next day, cells were washed thrice with PBS, incubated with an AlexaFluor 647 anti-rabbit (ab150079, Abcam, Cambridge, UK) and AlexaFluor 488 anti-mouse secondary antibodies (ab150113, Abcam, Cambridge, UK) diluted 1:200 dilutions in 1% BSA solution for 1 h at room temperature. Cells were washed thrice in PBS (5 min per wash), and then cover slips were mounted on a glass slide using Vectashield antifade mounting medium with DAPI (H-1200-10, Vector Labs, Burlingame, CA, USA). Slides were visualized using a confocal microscope (Keyence Corporation, Osaka, Japan).

### 4.6. Microarray Analysis

Total RNA was extracted from the cells by the PureLink RNA Mini Kit (Invitrogen, Waltham, MA, USA). After RNA was qualified by Agilent 2100 Bioanalyzer, Cy-3-labeled cRNA was synthesized from 50 ng of total RNA using the Low Input Quick-Amp Labeling Kit, One-color (Agilent Technologies, Santa Clara, CA, USA) and purified using an RNeasy Mini Kit (Qiagen). The concentration of amplified cRNA and dye incorporation was quantified using NanoDropOne spectrophotometer (ND-ONE-W, Thermo Fischer Scientific, Waltham, MA, USA) and hybridized to SurePrint G3 Human Gene Expression v3 8 *×* 60K Microarray Kit (Design ID:072363, Agilent Technologies). After hybridization, arrays were washed consecutively by using Gene Expression Wash Pack (Agilent Technologies). Fluorescence images of the hybridized arrays were scanned using the SureScan Microarray Scanner (Agilent Technologies), and the scanned data were extracted with Feature Extraction software ver. 12.1.1.1 (Agilent Technologies, Santa Clara, CA, USA). The raw microarray data are deposited in the National Center for Biotechnology Information Gene Expression Omnibus (GEO Series GSE85226). Gene expression analysis was carried out using GeneSpring GX 14.9.1 (Agilent Technologies). Each measurement was divided by the 75th percentile of all measurements in that sample at per chip normalization. The genes filtrated by flags, detected in all samples, were subjected to further analyses.

### 4.7. Flow Cytometry Analysis

Human NPs (hNPs), cultured in DMEM supplemented with 10% FBS and 1% antibiotics, and cells isolated from alginate beads were analyzed using a FACSCanto II (BD, Franklin Lakes, NJ, USA). Approximately 2.5 × 10^4^ cells were resuspended in 50uL FACS buffer with antibodies. Only living cells were detected using a 7AAD (7AAD-400T, immunostep, Salamanca, Spain, 1:100) negative living gate. Cells were stained with FITC conjugatedanti human CD24 mAb (130-095-952, Miltenyi Biotec, Bergisch Gladbach, Germany, 1:1000), anti-human GD2 mAb (sc-53831, Santa Cruz, Dallas, TX, USA, 1:1000), anti-human TEK mAb (ab221154, abcam, Cambridge, UK, 1:1000), Alexa Fluor 647 goat anti-rabbit IgG secondary antibody (ab150079; abcam, 1:2500), and PE goat anti-mouse IgG secondary antibody (ab97024; abcam, 1:1000). FITC mouse IgG Isotype Control (551954; BD Biosciences, 1:1000). A PE-labeled mouse IgG Isotype Control (551436; BD Biosciences, 1:1000) and a rabbit IgG monoclonal Isotype Control (ab172730; abcam, 1:1000) served as negative controls. Staining was performed according to the manufacturer’s procedure, and results were analyzed using FACS Diva software (BD, Franklin Lakes, NJ, USA).

### 4.8. Statistical Analysis

Data were compared using the Mann–Whitney test and Student *t*-test using JMP version 9 (SAS Institute Inc., Cary, NC, USA), and *p* < 0.05 indicates a significant difference between groups.

## 5. Conclusions

Ectopic expression of *MYC*, *NOTO*, *SOX5*, *SOX6*, and *SOX9* in NHDFs cultured with alginate beads induced differentiation to an NP-like phenotype. Alginate beads and TGF-β signaling may be required for directly differentiating and reprogramming NHDFs to NP-like cells in vitro. Considering potential targets of sources for cell therapy, these findings will likely contribute to effective treatments for IVD diseases.

## Figures and Tables

**Figure 1 ijms-23-04059-f001:**
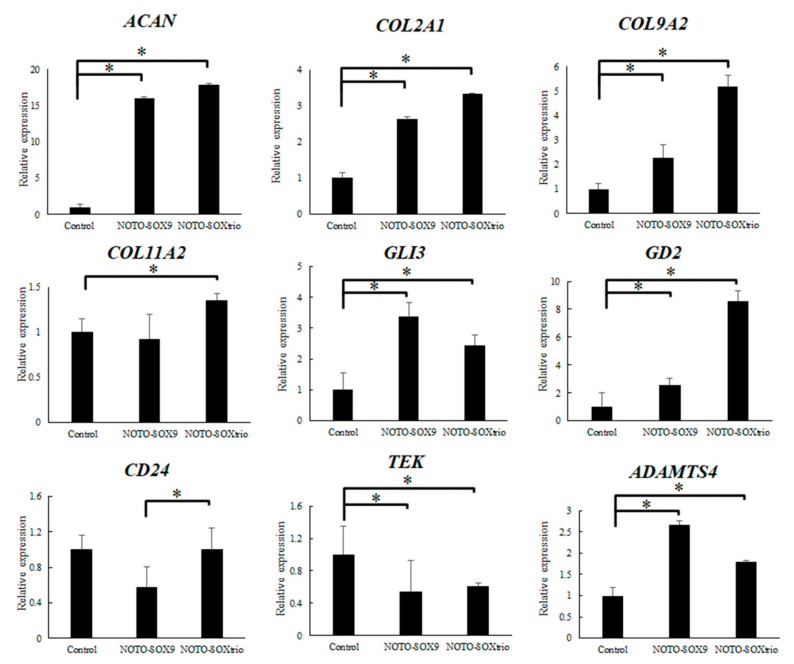
Transcriptional profiling of NHDFs. Error bars denote the mean ± SD. * *p* < 0.05 (Mann–Whitney test).

**Figure 2 ijms-23-04059-f002:**
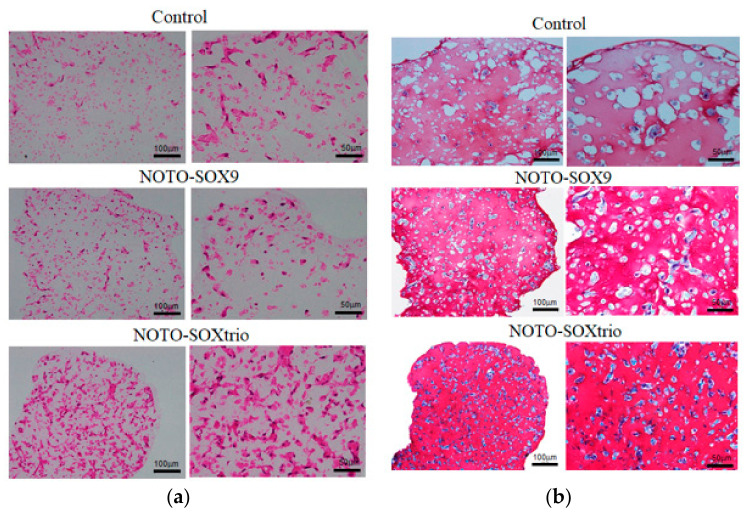
Histological analysis of NHDFs. NHDFs were cultured for 4 weeks in a chromogenic medium containing alginate beads: (**a**) HE-staining; (**b**) Safranin-O staining. Control indicates empty pcDNA3.1 transfected NHDFs.

**Figure 3 ijms-23-04059-f003:**
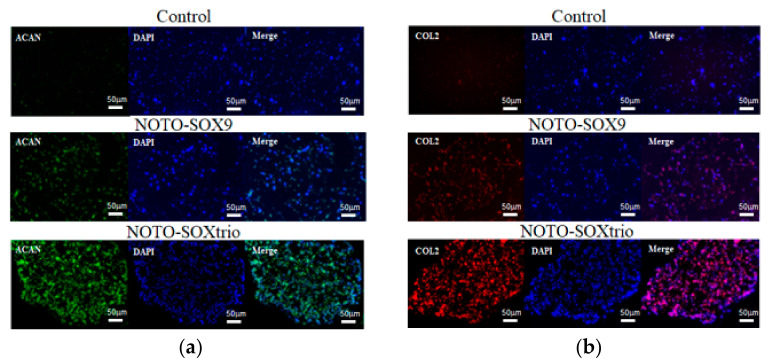
Immunofluorescence analysis of Aggrecan and type II collagen expression in NHDFs cultured with alginate beads. Paraffin-embedded, fixed sections were subjected to immunohistochemical analysis using antibodies specific for ACAN and type II collagen: (**a**) ACAN; (**b**) COL2. Control indicates empty pcDNA3.1 transfected NHDFs.

**Figure 4 ijms-23-04059-f004:**
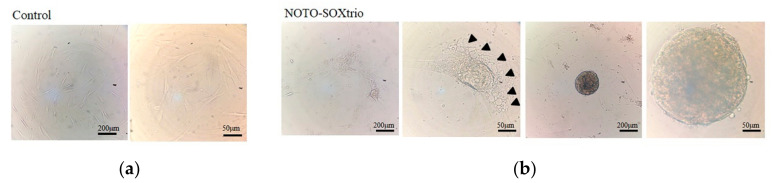
Cultures of the NOTO-SOXtrio group include vacuolated cells and CFU-S. The NOTO-SOXtrio group was cultured with alginate beads for 4 weeks: (**a**) control group; (**b**) NOTO-SOXtrio group. Arrowheads indicate intracellular vacuoles, and numerous intracellular vesicles were observed.

**Figure 5 ijms-23-04059-f005:**
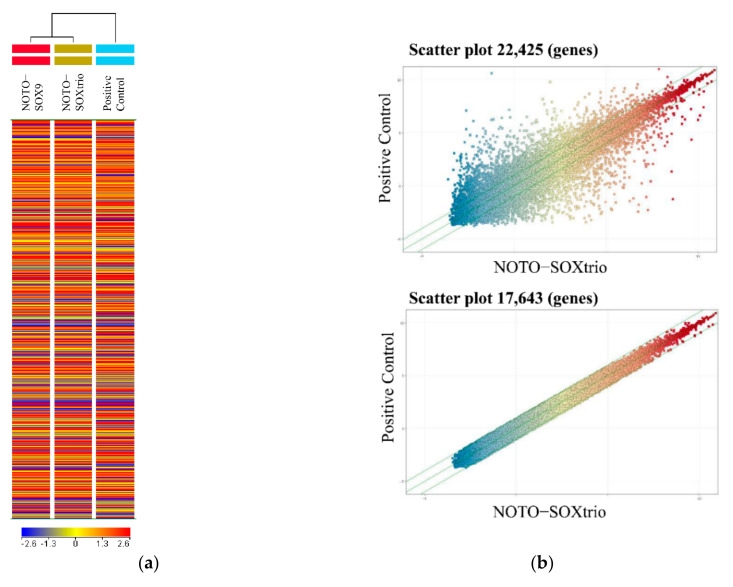
Microarray analysis of genes expressed by NHDFs: (**a**) The heatmap of the genes filtrated by flags (22,425 genes) was shown. All groups showed mean expression levels (each group; n = 3). (**b**) Scatter plot analysis. The lower panels show the number of genes with expression levels differing 2-fold compared with the positive control (NP) and the NOTO-SOXtrio group. (**c**) Heatmap of NP-specific genes (Positive control, n = 3; NOTO-SOXtrio, n = 3). (**d**) Scatter plot of NP-specific genes (n = 54).

**Figure 6 ijms-23-04059-f006:**
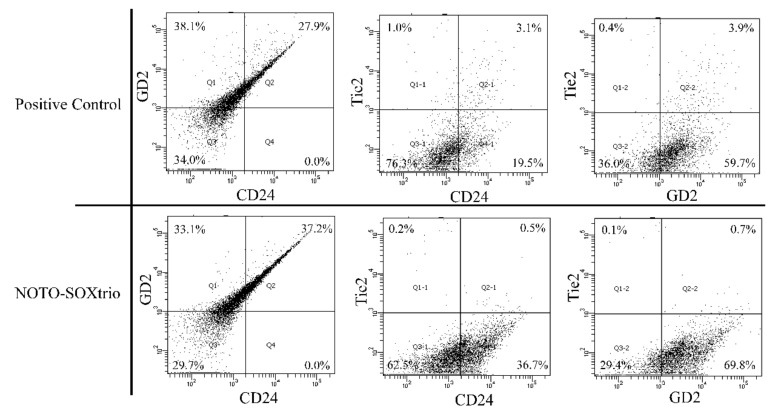
FACS analysis of NP-like cells.

**Figure 7 ijms-23-04059-f007:**
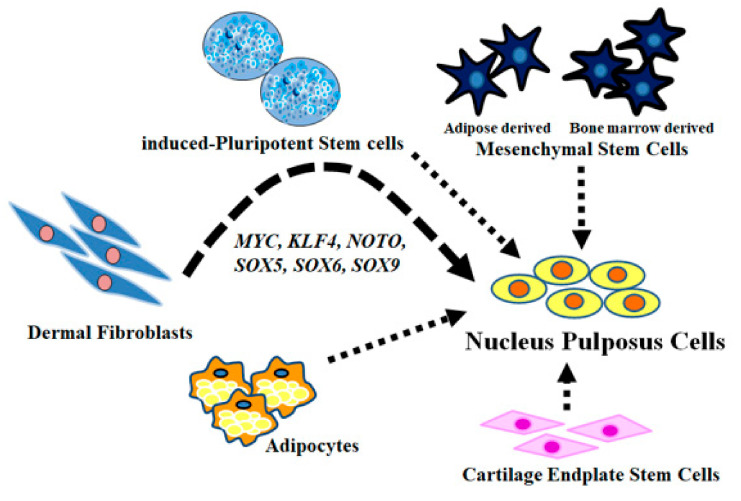
Potential targets of cell sources.

## Data Availability

The data presented in this study are available on request from the corresponding author.

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
