# Peer review of "Direct Reprogramming and Induction of Human Dermal Fibroblasts to Differentiate into iPS-Derived Nucleus Pulposus-like Cells in 3D Culture"

_ijms, 2022, doi:10.3390/ijms23074059_

Round 1
Reviewer 1 Report
The authors present a timely original work on how human dermal fibroblasts can be reprogrammed to stem cells, i.e. iPSC induced pluripotent stem cells.
My question is on the title whether this is appropriate. The question is whether the protocol of the authors is a protocol to generate iPSC? This keyword should come either in the title as well and at least in the keywords.
I recommend adding a table with abbreviations at the end as there are so many in this manuscript.
The authors cite correctly the recent works by Columbier et al. on how NP-like cells can be generated by overexpression of NOTO.
My comments are on the section of the FACS analysis:
The authors should also provide how much antibody they used per M of cells for the staining. This could affect reproducibility if this is repeated by other labs.
Figure 1 The gene of Tie2 (CD202b) is called the TEK gene. So, this should be changed in this figure as the bar plots refer to RNA transcript expression.
Figure 5 is too compacted. As a reader, I hardly can get anything out as the labels are too small. I only see some nice colours. So, to give it a scientific value the authors should split this figure into 2 figures and enlarge the figures considerably or put one figure as a supplementary figure where a high-quality figure can be downloaded.
Figure 6 appeared very blurry in my manuscript version. Furthermore, possibly the graphics would profit from the usage of different colours. Also, the font size should be enlarged on the labels of the axes.
General comment on gene nomenclature: Usually, human genes are not in italics: Please, consult this guideline, e.g. ACAN if the protein is meant and Acan if mRNA transcript is meant. There are some inconsistencies in the discussion.
https://www.ncbi.nlm.nih.gov/pmc/articles/PMC7494048/
The references are fine and the authors cited the most important literature.
Author Response
Review Report Form (Reviewer 1)
Point-by-Point Responses
Comments and Suggestions for Authors
The authors present a timely original work on how human dermal fibroblasts can be reprogrammed to stem cells, i.e. iPSC induced pluripotent stem cells.
My question is on the title whether this is appropriate. The question is whether the protocol of the authors is a protocol to generate iPSC? This keyword should come either in the title as well and at least in the keywords.
Response: Thank you for your comment. As you suggested, we think it better that the term of “iPSC” should come either in the title as well and at least in the keywords. Therefore, we rewrote the title " Direct Reprogramming and Induction of Human Dermal Fibroblasts to Differentiate into iPS-derived Nucleus Pulposus-like Cells in 3D Culture", and added the keyword “induced pluripotent stem cells”.
I recommend adding a table with abbreviations at the end as there are so many in this manuscript.
Response: Thank you for your comment. As you suggested, we added the abbreviations at the end of this manuscripts, Lines 453-494 as follows.
Abbreviations
ACAN aggrecan
ADAMTS4 ADAM metallopeptidase with thrombospondin type 1 motif 4
AF annulus fibrosus
BSA bovine serum albumin
CD24 CD24 molecule
CFU-F colony-forming units-fibroblastic
CFU-S colony-forming units-spherical
COL2A1 collagen type II alpha 1 chain
COL9A2 collagen type IX alpha 2 chain
COL11A2 collagen type XI alpha 2 chain
DAPI 4',6-diamidino-2-phenylindole
DMEM Dulbecco's Modified Eagle Medium
ECM extracellular matrix
FBS fetal bovine serum
FACS fluorescence-activated cell sorting
FOXA2 forkhead box A2
GAG glycosaminoglycan
GAPDH glyceraldehyde-3-phosphate dehydrogenase
GD2 disialoganglioside GD2
GLI3 GLI family zinc finger 3
HE hematoxylin and eosin
IGF insulin-like growth factor
IVD intervertebral disc
KLF4 kruppel like factor 4
NHDFs normal human dermal fibroblasts
NODAL nodal growth differentiation factor
NOTO notochord homeobox
NP nucleus pulposus
MYC MYC proto-oncogene, bHLH transcription factor
PBS phosphate-buffered saline
qPCR quantitative polymerase chain reaction
SHH sonic hedgehog signaling molecule
SOX2 SRY (sex-determining region Y)-box 2
SOX5 SRY (sex-determining region Y)-box 5
SOX6 SRY (sex-determining region Y)-box 6
SOX9 SRY (sex-determining region Y)-box 9
SOXtrio SOX5, SOX6 and SOX9
T brachyury, T-box transcription factor T
TEK TEK receptor tyrosine kinase
TGF-beta1 transforming growth factor beta 1
The authors cite correctly the recent works by Columbier et al. on how NP-like cells can be generated by overexpression of NOTO.
Response: Thank you for your comment.
My comments are on the section of the FACS analysis:
The authors should also provide how much antibody they used per M of cells for the staining. This could affect reproducibility if this is repeated by other labs.
Response: Thank you for your comment. As you suggested, we provided about the diluted concentration of the antibody used for the staining in section 4.7, Lines 407-423.
Figure 1 The gene of Tie2 (CD202b) is called the TEK gene. So, this should be changed in this figure as the bar plots refer to RNA transcript expression.
Response: Thank you for your comment. As you suggested, we changed from “Tie2” to “TEK” in Figure 1.
Figure 5 is too compacted. As a reader, I hardly can get anything out as the labels are too small. I only see some nice colours. So, to give it a scientific value the authors should split this figure into 2 figures and enlarge the figures considerably or put one figure as a supplementary figure where a high-quality figure can be downloaded.
Response: Thank you for your comment. As you suggested, we made significant changes to Figure 5 for giving a scientific value. We split this figure into 2 figures and enlarge the figures considerably.
Figure 6 appeared very blurry in my manuscript version. Furthermore, possibly the graphics would profit from the usage of different colours. Also, the font size should be enlarged on the labels of the axes.
Response: Thank you for your comment. As you mentioned, Figure 6 might appear very blurry. Therefore, we enlarged the image and font size of the labels of the axes in Figure 6. We tried to use different colours for the two types of expression, but the software (FACS Diva software) we are using has limitations in the gradation and overlay function, and it is not possible to dye them clearly, so we decided to use a single color for the figure.
General comment on gene nomenclature: Usually, human genes are not in italics: Please, consult this guideline, e.g. ACAN if the protein is meant and Acan if mRNA transcript is meant. There are some inconsistencies in the discussion.
Response: Thank you for your comment. According to the guide line, the HGNC endorses the use of italics to denote genes, alleles and RNAs to distinguish them from proteins in the section “Gene symbol usage”. Therefore, we used the italics for the human genes. In case of the genes of rodent, the initial letter of gene was written in a capital letter like as “Acan”. As you suggested, our paper contained some inconsistencies in discussion section, therefore, we rewrote it in the discussion section.
https://www.ncbi.nlm.nih.gov/pmc/articles/PMC7494048/
The references are fine and the authors cited the most important literature.
Response: Thank you again.

Reviewer 2 Report
Dear authors: Thank you for your manuscript detailing reprogramming of human dermal fibroblasts. The data seems largely sound, if not terrible innovative since fibroblasts have been shown in many reports to be able to be reprogrammed-including into disc cells. It should be noted when you refer to NP cells which phenotype of NP you are referring to since mature IVDs (such a bovine, human) do not contain highly vacuolated cells that are more the situation with young humans, non-chondrodystrophic canines, rabbits. Therefore it is not certain to what you refer in these cases.
You suggest that this approach might be useful to treat IVD DDD-how do you anticipate such an approach being translatable when there is no evidence that transplanted cells survive and engraft into the host tissue within the inhospitable degenerative disc environment? You should entertain this challenge and potential answers within your discussion.
Author Response
Review Report Form (Reviewer 2)
Point-by-Point Responses
Comments and Suggestions for Authors
Dear authors: Thank you for your manuscript detailing reprogramming of human dermal fibroblasts. The data seems largely sound, if not terrible innovative since fibroblasts have been shown in many reports to be able to be reprogrammed-including into disc cells. It should be noted when you refer to NP cells which phenotype of NP you are referring to since mature IVDs (such a bovine, human) do not contain highly vacuolated cells that are more the situation with young humans, non-chondrodystrophic canines, rabbits. Therefore it is not certain to what you refer in these cases.
Response: Thank you for your comment. As you suggested, adult-human mature IVD cells may do not contain highly vacuolated cells that are more the situation with young humans, non-chondrodystrophic canines, rabbits. Our data showed that the vacuolated cells were observed in the process of direct reprogramming and differentiation from human dermal fibroblasts into NP-like cells. Therefore, it may mean the evidence that the NP-like cells have returned to younger cells due to their reprogramming and differentiation.
You suggest that this approach might be useful to treat IVD DDD-how do you anticipate such an approach being translatable when there is no evidence that transplanted cells survive and engraft into the host tissue within the inhospitable degenerative disc environment? You should entertain this challenge and potential answers within your discussion.
Response: Thank you for your question. As you suggested, there may be no evidence that transplanted cells survive and engraft into the host tissue within the inhospitable degenerative disc environment. The cellular transplantation and engraftation techniques are needed in the future. An essential consideration of cell transplantation is the carrier or medium in which cells are implanted. Carriers used within these clinical trials are collagen sponge, fibrin, hyaluronic acid, and hyaluronic acid derivative gel. Gels and scaffold can enhance in situ incorporation and guide differentiation. Encapsulation is also a potential method to limit cell leakage. Therefore, we added to the discussion section, lines 262-279 as follows.
“Finally, there is no evidence that this transplanted NP-like cells survive and engraft into the host tissue within the inhospitable degenerative disc environment. Bio-material scaffolds can provide mechanical and structural support in the regeneration of IVDs and may also serve as a carrier for therapeutic cell delivery. An essential consideration of cell therapy technique is the scaffolds in which cells are implanted. Although various scaffolds have been tried in clinical or animal models as collagen sponge [47], collagen microspheres [48], fibrin [49], hyaluronic acid hydrogels [50] and alginate beads [51], it has not reached the general treatment for IVD diseases. Gels and scaffolds can enhance in situ incorporation and guide differentiation. Scaffolds itself is a potential method to encapsulation preventing cell leakage. Further cellular transplantation and engraftation techniques are needed in the future.”
47). Yoshikawa, T.; Ueda, Y.; Miyazaki, K.; Koizumi, M.; Takakura, Y. Disc regeneration therapy using marrow mesenchymal cell transplantation: a report of two case studies. Spine 2010, 35, E475-80. doi: 10.1097/BRS.0b013e3181cd2cf4.
48). Li, Y.Y.; Diao, H.J.; Chik, T.K.; Chow, C.T.; An, X.M.; Leung, V.; Cheung, K.M.; Chan, B.P. Delivering mesenchymal stem cells in collagen microsphere carriers to rabbit degenerative disc: reduced risk of osteophyte formation. Tissue Eng. Part A. 2014, 20, 1379-1391. doi: 10.1089/ten.TEA.2013.0498.
49). Coric, D.; Pettine, K.; Sumich, A.; Boltes, M.O. Prospective study of disc repair with allogeneic chondrocytes presented at the 2012 Joint Spine Section Meeting. J. Neurosurg. Spine 2013, 18, 85-95. doi: 10.3171/2012.10.SPINE12512.
50). Kim, D.H.; Martin, J.T.; Elliott, D.M.; Smith, L.J.; Mauck, R.L. Phenotypic stability, matrix elaboration and functional matura-tion of nucleus pulposus cells encapsulated in photocrosslinkable hyaluronic acid hydrogels. Acta Biomater. 2015, 12, 21-29. doi: 10.1016/j.actbio.2014.10.030.
51). Wang, H.; Zhou, Y.; Huang, B.; Liu, L.T.; Liu, M.H.; Wang, J.; Li, C.Q.; Zhang, Z.F.; Chu, T.W.; Xiong, C.J. Utilization of stem cells in alginate for nucleus pulposus tissue engineering. Tissue Eng. Part A. 2014, 20, 908-920. doi: 10.1089/ten.TEA.2012.0703.

Reviewer 3 Report
The artice is nicely written and interesting and suitable for publication. I have some qestions below:
- Introduction, line 45: authors write about NP and AF, can tehy sey a few sentences about cells of the terminal plate? They are equally , if not more important in the disc structure and degeneration.
- part 2.1: the description of Figure 1 is missing. The comment on COL112, CD24, ADAMTS4 is missing.
- line 94: how are the authors sure that the small round cells were chondrocytes after safranin staining?
- line 103. Safranin staining must be wrong here. Did the authors really use safranin or is this a mistake? They probably ment antibodies and immunofluorescence here. Please clarify.
- Figure 3: the controls must also express ACAN and COL2. Not evident from the figure. This may be a sign that the dedifferentiation of the NP cells might occur, since these cells do not express these two markers after dedifferentiation. Have the authors used the right cells?
- line 111: vacuolated
- section 2.5: how many patients did the authros include? A single patient with scoliosis or more? Are these patients appropriate for control? The disc must not be degenerated here. Please clarify.
- line 278: bovine
- section 4.3: how can be the authors sure that the cells for control were not dedifferentiated (the change of phenotype)? With the NP cells, this may occur even in the primary culture.
- How many cells did the authors get during the isolation? How many times were they subcultured? What passage of the isoated cells was used for the experiment? Are the authors therefore sure that the cells for control were ok? Nothing is written about COL2 and ACAN here. The cells may loose the phenotype during the subcultivation. For 6 million cells, a few subcultivations are needed, unless you ahve a lot of tissue. And during the subcultivation, the NP cells change. Please clarify this part.
- line 316: what antibodies exactly: antirabbit, antimouse?
- line 319: diluted
- Conclusions can be improved- expanded.
Author Response
Review Report Form (Reviewer 3)
Point-by-Point Responses
Comments and Suggestions for Authors
The artice is nicely written and interesting and suitable for publication. I have some qestions below:
Response: Thank you for your acceptability comment.
Introduction, line 45: authors write about NP and AF, can they sey a few sentences about cells of the terminal plate? They are equally, if not more important in the disc structure and degeneration.
Response: Thank you for your question. As you suggested, the structure of the terminal plate is complicated and important for degenerative process. We added the sentences about cells of terminal plate as follows in lines 47-52.
“The IVD comprises the nucleus pulposus (NP), the annulus fibrosus (AF) and endplate. An endplate of IVD is the transition region where a vertebral body and intervertebral disc interface with each other, and the endplate is divided into two parts, cartilaginous and bony endplate. The cartilaginous endplate contains cartilage-like cells and helps maintain the form and function of the disc. “
part 2.1: the description of Figure 1 is missing. The comment on COL112, CD24, ADAMTS4 is missing.
Response: Thank you for your comments. We added the comments on COL11A2, CD24, and ADMTS4 in Results section, Lines 92-95, as follow.
“COL11A2 was also significantly increased in NOTO-SOXtrio groups. At the same time, ADAMTS4 was significantly increased in the NOTO-SOX9 and NOTO-SOXtrio groups. CD24 was significantly increased in NOTO-SOXtrio groups compared with that of NOTO-SOX9 group.”
line 94: how are the authors sure that the small round cells were chondrocytes after safranin staining?
Response: Thank you for your comments. Regarding whether the small round cells after safranin-O staining were chondrocytes, it was confirmed that they were chondrocytes because the expression of COL2 and aggrecan was abundant by fluorescent immunostaining as shown in Figure 3.
line 103. Safranin staining must be wrong here. Did the authors really use safranin or is this a mistake? They probably ment antibodies and immunofluorescence here. Please clarify.
Response: Thank you for your comments. As you suggested, this is a mistake. Therefore, we rewrote it in Lines 117-118 as follows.
“Immunofluorescence revealed that Aggrecan and Type II collagen, which are characteristic of the NP, were abundantly expressed in marker-gene transfected NHDFs but not in control transfectants.”
Antibodies for immunofluorescence was added in Methods section, Lines 372-382 as follows.
“For immunofluorescence analysis, the deparaffinized sections were permeabilized with 0.1% Triton X-100 (065K0122, Merk, MA, USA) for 30 min, washed twice with PBS, and then incubated for 1 h at room temperature in 3% BSA in PBS (blocking solution). The cells were then incubated, overnight at 4 ºC with rabbit polyclonal anti-type II collagen (ab34712, Abcam, Cambridge, UK) and mouse monoclonal anti-aggrecan (sc-33695, Santa Cruz, TX, USA) antibodies (diluted 1:50 in 1% BSA solution). The next day, cells were washed thrice with PBS, incubated with an AlexaFluor 647 anti-rabbit (ab150079, Abcam, Cambridge, UK) and AlexaFluor 488 anti-mouse secondary antibodies (ab150113, Abcam, Cambridge, UK) diluted 1:200 dilutions in 1% BSA solution for 1 h at room temperature. Cells were washed thrice in PBS (5 min per wash) and then cover slips were mounted on a glass slide using Vectashield antifade mounting medium with DAPI (H-1200-10, Vector Labs, CA, USA). Slides were visualized using a confocal microscope (Keyence Corporation, Osaka, Japan).”
Figure 3: the controls must also express ACAN and COL2. Not evident from the figure. This may be a sign that the dedifferentiation of the NP cells might occur, since these cells do not express these two markers after dedifferentiation. Have the authors used the right cells?
Response: Thank you for your comments. The controls indicate human dermal fibroblasts (NHDFs). Therefore, these cells expressed very little COL2 and AGC. As confused with the positive controls in Figure 5 and 6, we added that control indicates empty pcDNA3.1 transfected NHDFs in Figure 2 and 3 legends.
line 111: vacuolated
Response: Thank you for your comments. We rewrote it.
section 2.5: how many patients did the authros include? A single patient with scoliosis or more? Are these patients appropriate for control? The disc must not be degenerated here. Please clarify.
Response: Thank you for your question. Three scoliotic patients were included in this study. The intervertebral discs of 6-8 years old boys with congenital scoliosis were removed and used for this study. All patients had nothing of IVD degeneration confirmed by MRI. We added it in Material and Methods section of the manuscripts, Lines 328-330.
line 278: bovine
Response: Thank you for your comments. We rewrote it.
section 4.3: how can be the authors sure that the cells for control were not dedifferentiated (the change of phenotype)? With the NP cells, this may occur even in the primary culture.
Response: Thank you for your comments. As you mentioned, we have ever observed in dedifferentiation of NP cells of the primary culture. We have been basic researches of NP cells or intervertebral disc1)-4). It has been confirmed that even if dedifferentiated once, it redifferentiates for several weeks after culturing with alginate beads and chondrocyte medium. The expression levels of type II collagen and aggrecan are important for whether or not they can be redifferentiated, and as mentioned in Figure 4, we believe that the presence of vacuolated cells and colony formation is evidence of redifferentiation.
- A functional SNP in CILP, encoding cartilage intermediate layer protein, is associated with susceptibility to lumbar disc disease. Seki S, Kawaguchi Y, Chiba K, Mikami Y, Kizawa H, Oya T, Mio F, Mori M, Miyamoto Y, Masuda I, Tsunoda T, Kamata M, Kubo T, Toyama Y, Kimura T, Nakamura Y, Ikegawa S. Nat Genet. 2005 Jun;37(6):607-12. doi: 10.1038/ng1557. Epub 2005 May 1.
- Effect of small interference RNA (siRNA) for ADAMTS5 on intervertebral disc degeneration in the rabbit anular needle-puncture model. Seki S, Asanuma-Abe Y, Masuda K, Kawaguchi Y, Asanuma K, Muehleman C, Iwai A, Kimura T. Arthritis Res Ther. 2009;11(6):R166. doi: 10.1186/ar2851. Epub 2009 Nov 4.
- Lumbar disc degeneration progression in young women in their 20's: A prospective ten-year follow up. Makino H, Kawaguchi Y, Seki S, Nakano M, Yasuda T, Suzuki K, Ikegawa S, Kimura T. J Orthop Sci. 2017 Jul;22(4):635-640. doi: 10.1016/j.jos.2017.03.015. Epub 2017 Apr 18. PMID: 28431805
- A selective inhibition of c-Fos/activator protein-1 as a potential therapeutic target for intervertebral disc degeneration and associated pain. Makino H, Seki S, Yahara Y, Shiozawa S, Aikawa Y, Motomura H, Nogami M, Watanabe K, Sainoh T, Ito H, Tsumaki N, Kawaguchi Y, Yamazaki M, Kimura T. Sci Rep. 2017 Dec 5;7(1):16983. doi: 10.1038/s41598-017-17289-y.
How many cells did the authors get during the isolation?
Response: Thank you for your questions. Depending on the amount of tissue, excision of two intervertebral discs at the age of 6-8 years will give about 2 x 105 ~ NP cells in our experience.
How many times were they subcultured?
Response: Thank you for your questions. NP cells were subcultured 2 or 3 times to increase the number of cells.
What passage of the isoated cells was used for the experiment?
Response: Thank you for your questions. We try to keep the cells used for the experiment within 3 passages in total, including subculture.
Are the authors therefore sure that the cells for control were ok? Nothing is written about COL2 and ACAN here. The cells may loose the phenotype during the subcultivation. For 6 million cells, a few subcultivations are needed, unless you ahve a lot of tissue. And during the subcultivation, the NP cells change. Please clarify this part.
Response: Thank you for your comments and questions. As you suggested, we added the subcultivations and confirmations of cell phenotype in Methods section, Lines 351-355 as follows.
“NP cells were subcultured 2 or 3 times to increase the number of cells. We try to keep the cells used for the experiment within 3 passages in total, including subculture. If the phenotype of NP cell was dedifferentiated within 3 passages, the cells were basically destroyed or redifferentiated with alginate beads with Chondrocyte Differentiation Medium (CC-3225; Lonza, Basel, Switzerland) for experimental study. “
line 316: what antibodies exactly: antirabbit, antimouse?
Response: Thank you for your comments. We added it as follows.
rabbit polyclonal anti-type II collagen (ab34712, Abcam, Cambridge, UK)
mouse monoclonal anti-aggrecan (sc-33695, Santa Cruz, TX, USA)
line 319: diluted
Response: Thank you for your comments. We rewrote it.
Conclusions can be improved- expanded.
Response: Thank you for your comments. As you suggested, we rewrote in Conclusion, Lines 430-434 as follows.
“Ectopic expression of MYC, NOTO, SOX5, SOX6, and SOX9 in NHDFs cultured with alginate beads induced differentiation to an NP-like phenotype. Alginate beads and TGF-β signaling may be required for directly differentiating and reprogramming NHDFs to NP-like cells in vitro. Considering potential targets of sources for cell therapy, these findings will likely contribute to effective treatments for IVD diseases.”

Round 2
Reviewer 2 Report
Dear authors thank you for providing the revised manuscript. Generally the paper reads better and the revised text is a big improvement. I still think that the tissue culture experiments ought to have been performed in a hypoxic tissue compartment in order to better mimic the milieu of the IVD, however with a sentence to this effect in a 'limitations' section of the paper I think that it is acceptable.
Author Response
Point-by-Point Responses
Comments and Suggestions for Authors
Dear authors thank you for providing the revised manuscript. Generally the paper reads better and the revised text is a big improvement. I still think that the tissue culture experiments ought to have been performed in a hypoxic tissue compartment in order to better mimic the milieu of the IVD, however with a sentence to this effect in a 'limitations' section of the paper I think that it is acceptable.
Response
Thank you for your comments. As you suggested, we might need to perform that the IVD tissue culture experiments ought to have been performed in a hypoxic tissue compartment in order to better mimic the milieu of the IVD. Therefore, we added a limitation in the Discussion section, lines 251-255 as follows.
“On the other hand, highly mature and undegenerated human NP cells may not have so many vacuolated and colony-forming cells. Although we believe that the NP cells of the 3D culture was able to maintain the phenotype, it may have been better to perform the culture in a hypoxic compartment in order to better mimic the milieu of the IVD. This is our limitation in this study.”
Thank you, again.
